# Distribution of Hepatitis B prevention services in Wakiso District, Central Uganda

**Tonny Ssekamatte**[1]*, **John Bosco Isunju**[1], **Aisha Nalugya**[1], **Solomon Tsebeni Wafula**[1], **Rebecca Nuwematsiko**[1], **Doreen Nakalembe**[1], **Winnifred K. Kansiime**[1], **Naume Muyanga**[1], **Joana Nakiggala**[1], **Justine N. Bukenya**[2], **Richard K. Mugambe**[1]

**1** Department of Disease Control and Environmental Health, School of Public Health, College of Health Science, Makerere University, Kampala, Uganda, **2** Department of Community Health and Behavioural Sciences, School of Public Health, College of Health Science, Makerere University, Kampala, Uganda

* ssekamattet.toca@gmail.com, tssekamatte@musph.ac.ug

## Abstract

Hepatitis B Virus (HBV) infection remains a significant global public health challenge especially in low-and-middle income countries. Although there are significant global and national efforts to control Hepatitis B, equitable distribution and access to prevention services such as testing and vaccination remains a challenge. Efforts to increase access are hindered by inadequate evidence on the availability and distribution of HBV services. This cross-sectional study aimed at generating evidence of the distribution of HBV prevention services in Wakiso District, Uganda. A total of 55 healthcare facilities (HCFs) including 4 hospitals, and 51 primary care facilities were surveyed. Data were collected using an electronic structured questionnaire and analysed using STATA 14.0. A chi-square test was performed to establish the relationship between HCF characteristics and the availability of hepatitis B services. ArcGIS (version 10.1) was used to generate maps to illustrate the distribution of hepatitis B prevention services. We found out that the hepatitis B vaccine was available in only 27.3% (15) of the HCF, and 60% (33) had testing services. Receipt of the hepatitis B vaccine doses in the last 12 months was associated with the level (p = ≤0.001) and location (p = 0.030) of HCF. Availability of the hepatitis B vaccine at the time of the survey was associated with the level (p = 0.002) and location (p = 0.010) of HCF. The availability of hepatitis B testing services was associated with the level (p = 0.031), ownership (p≤0.001) and location (p = 0.010) of HCF. HCFs offering vaccination and testing services were mostly in urban areas, and close to Kampala, Uganda's capital. Based on this study, hepatitis B prevention services were sub-optimal across all HCF levels, locations, and ownership. There is a need to extend hepatitis B prevention services to rural, public and private-not-for-profit HCFs.

## Introduction

Hepatitis B virus (HBV) infection remains a global public health concern, affecting over 257 million people [1,2]. Low-and-middle-income countries (LMICs) bear the greatest burden of

**Funding:** The authors received no specific funding for this work.

**Competing interests:** The authors have declared that no competing interests exist.

HBV infection, accounting for almost all (96%) individuals living with the disease [1]. The incidence of HBV infection is 9.2 times higher in LMICs than high-income countries [1]. The WHO African region has the second highest prevalence of HBV infection (6.1%) while the Americas has the lowest (0.7%) [1]. The prevalence of HBV infection in Uganda is 4.1% among adults and 0.6% among children. It is highest in the mid-north (4.6%) and lowest in the south-western region (0.8%) of the country [3].

HBV infection affects the liver, and is transmitted through contact with blood or other body fluids of an infected person. A large proportion of HBV cases are carriers, who may remain asymptomatic for a long time, thus increasing their risk of liver cirrhosis, hepatocellular carcinoma, and early death [1]. The risk of chronic HBV infection and co-infection is high among under-fives, health care providers, and people living with HIV/AIDS [1,4,5]. The risk is also high among Ugandan adults born before 2002 (the year of the introduction of the HBV vaccine) since they missed out on the routine vaccination given to newborns at 6, 10, and 14 weeks of age through the expanded programme on immunisation. Despite the high public health burden of HBV infection in LMICs including Uganda, access to prevention services such as testing and vaccination remains suboptimal [1,6]. Evidence of the readiness and distribution of HBV prevention services is also limited. Yet, access to HBV prevention services is critical in contributing to the elimination of HBV infection by 2030, as stipulated in the 2016–2021 Global Health Sector Strategy (GHSS) on viral hepatitis [7].

Availability of testing services is key in the diagnosis of chronic HBV, and consequently linkage of patients to appropriate care and treatment, which eventually delays progression of liver disease [6–8]. Additionally, availability of testing services provides an avenue for counselling on risky behaviours, provision of prevention commodities such as sterile needles and syringes, and vaccination [6,8]. Vaccination reduces the risk of HBV infection, and co-infection with Hepatitis D virus, which is known to aggreveate the outcome of HBV infection [1,9,10]. Although evidence on global and regional coverage of hepatitis B vaccination services among adults is scarce, high-income countries with vaccination data reveal low rates [11]. Limited access to testing and vaccination services among adults born before the HBV vaccination era increases their risk of infection and death [1,6]. Available data on vaccination coverage among groups at an elevated risk of the HBV infection in LMICs reveal suboptimal rates [12–15].

There have been various strategies at global, regional and national level aimed at reducing disparities in access to hepatitis B prevention, care and treatment services [16]. In 2016, the World Health Organisation (WHO) developed a global health sector strategy on viral hepatitis (2016–2021) which envisioned a world where viral hepatitis transmission is halted and everyone living with viral hepatitis has access to safe, affordable and effective prevention, care and treatment services [1,7]. In response to the WHO recommendations and targets, the Ugandan Ministry of Health (MoH) called for increment in massive community awareness of HBV, screening and vaccination of all susceptible persons, improving the supply chain for HBV related commodities and supplies, enrolment of patients into treatment, care, and support programs, integration of services and resource mobilisation and equipping healthcare providers (HCPs) with adequate knowledge on the management of HBV infection. More so, the government of Uganda allocated US$2,8 million in 2016 towards the procurement of vaccines, laboratory reagents, and antiviral drugs for the prevention and treatment of Hepatitis B [17].

Despite efforts to improve the prevention of HBV infection in Uganda, evidence of distribution of related services including vaccination and testing is limited [18]. Our study utilised the service availability and readiness assessment (SARA) framework to establish readiness (e.g., availability of testing kits, trained personnel and vaccine storage facilities), and distribution of hepatitis B prevention services such as vaccination, testing and treatment in selected healthcare facilities (HCFs) in Wakiso district, Uganda.

## Materials and methods

### Study setting

This study was conducted in Wakiso district, Central Uganda in July 2018. Wakiso District is located in the Central Region of Uganda and encircles Kampala, Uganda's capital (Fig 1). Wakiso district is estimated to have a population of 3,519,300 people, of which 2,832,800 live in urban areas [19], making it the most populated district in Uganda. It has both an urban and rural setting, which provides an opportunity for comparison of HBV services across these settings. Wakiso district also has the highest number of HCFs making it an ideal site for the study. It has seven health sub-districts, with a total of 589 HCFs, among which 72 are owned and managed by the government (public), 477 are privately owned (private for-profit) and 40 are private not-for-profit [20]. Based on the level of the HCFs, 234 are private clinics, 153 are health centre IIs, 165 are health centre IIIs, 19 are health centre IVs, 14 are hospitals, and 3 are specialised clinics. Wakiso district is also home to the Entebbe regional referral hospital [20].

In terms of management of health services in Uganda, healthcare services in the district are overseen by the District Health Officer who is supported by two assistants (Assistant District Health Officers-Environmental Health, and Maternal and Child Health). The district is further divided into health sub-districts, which are overseen by a management committee. The functions of the committee include monitoring the general administration of the health Sub-district on behalf of the district local council and MoH. Each of these committees has a health sub-district manager, often an in-charge at a health centre IV, whose role involves coordination, planning, and budgeting for medical supplies and infrastructure, budget execution, and supervision of lower-level HCFs such as health centre IIIs and IIs [21].

### Study design and data collection tools

A cross-sectional study design was used to collect quantitative data from each of the study HCFs. An electronic structured questionnaire (S1 Text) was used to obtain data from the HCF managers. Where observations were required, questions prompting the researcher to make such observations were embedded in the data collection tool. The electronic structured questionnaire was developed by the lead investigators (TS, JBI, AN, and RKM) after a thorough review of literature on provision of vaccination services [22,23], and SARA framework [24,25]. The SARA framework is designed to systematically generate a set of core indicators of services, which can be used to measure progress in health system strengthening over time [24,26]. The study specifically focused on service availability, which is defined as the physical presence of the delivery of services, encompassing the health infrastructure, core health personnel, and service utilization [24,26]. The SARA framework is useful in the identification of the proportion of HCFs offering a particular intervention/service-in this case hepatitis B vaccination services, and whether the HCFs offering HBV services met the minimum standards in terms of equipment, trained staff and guidelines, diagnostic capacity, and medicines to provide an adequate level of service (service readiness) [24,27,28]. The data collection tool was reviewed and evaluated for face and internal validity by experts in viral hepatitis research, based at the Makerere University College of Health Sciences.

### Study variables

The validated tool captured data (S1 Data) on the HCF characteristics such as level (categorised as hospital, health centre IVs, and IIIs), ownership (Private for-profit (PFP), Private not-for-profit (PNFP), and Public/government-owned), and location (rural/urban). Urban HCFs included those administratively located in municipalities or town councils while rural HCFs

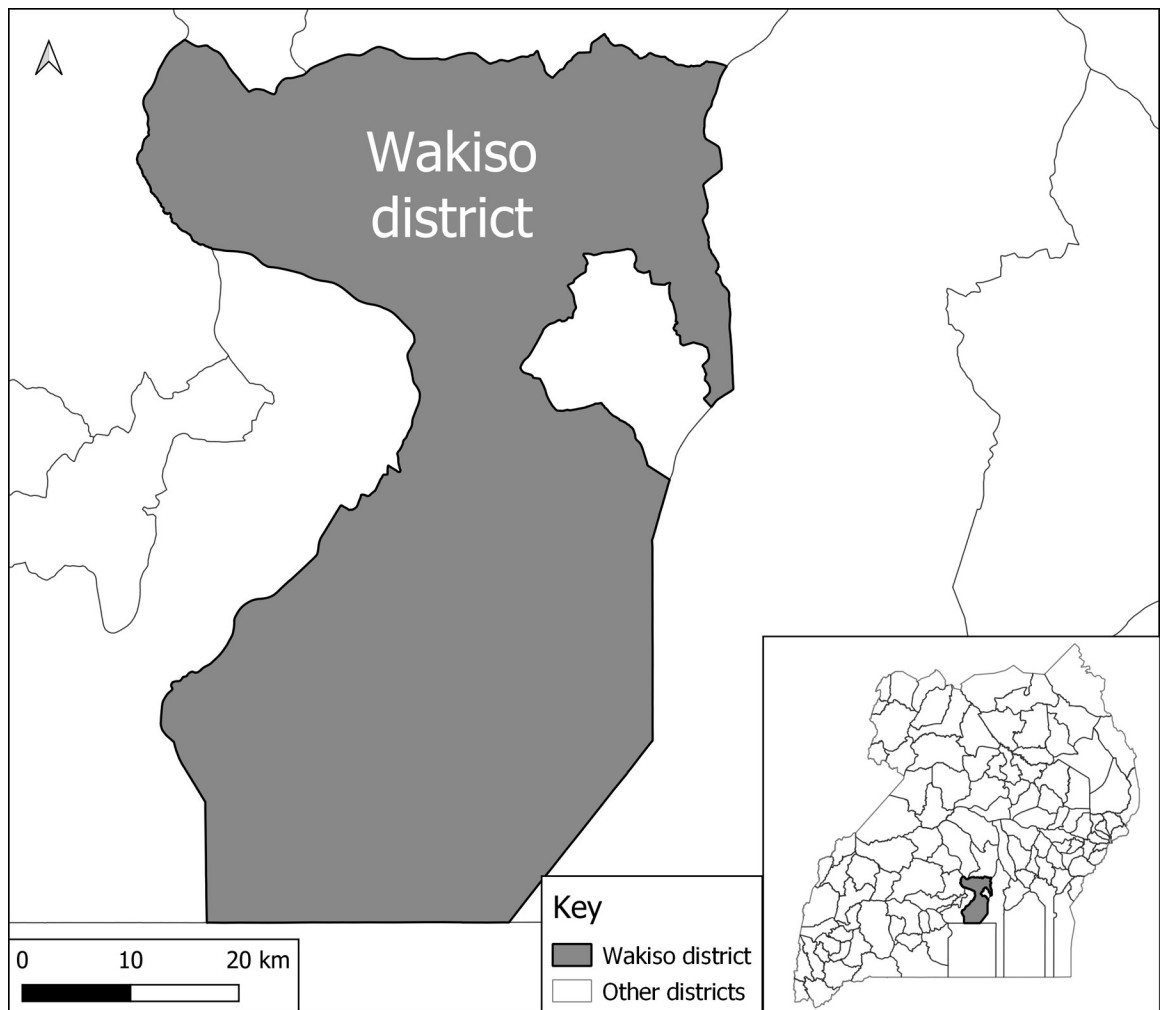

**Fig 1. Map showing the location of Wakiso district and its environs.**

were those located in a sub-county [29]. Data were collected on the availability of vaccination and testing services, trained personnel, and vaccination storage equipment such as refrigerators. In addition, observations were made to ascertain the availability of hepatitis B testing kits, vaccine doses, infection prevention and control promotion materials, and disposal of medical wastes such as sharps.

## Sample size, sampling procedure, data collection procedures, and data analysis

A total of 55 HCFs were surveyed (Fig 2). In order to be eligible for inclusion, a HCF had to provide high-risk medical interventions such as blood transfusions, delivering higher numbers of mothers, and other surgical procedures that can elevate the risk of transmission of HBV were selected [30,31]. We employed mixed sampling strategies in the selection of HCFs. With the aid of the national health facility inventory [20], we purposively selected all the general hospitals, Health centre IIIs, and IVs because they offered high-risk medical interventions such as caesarean deliveries and blood transfusion [12,31]. Given the level and nature of services provided, these HCFs were expected to offer hepatitis B vaccination services. In total, 4 general

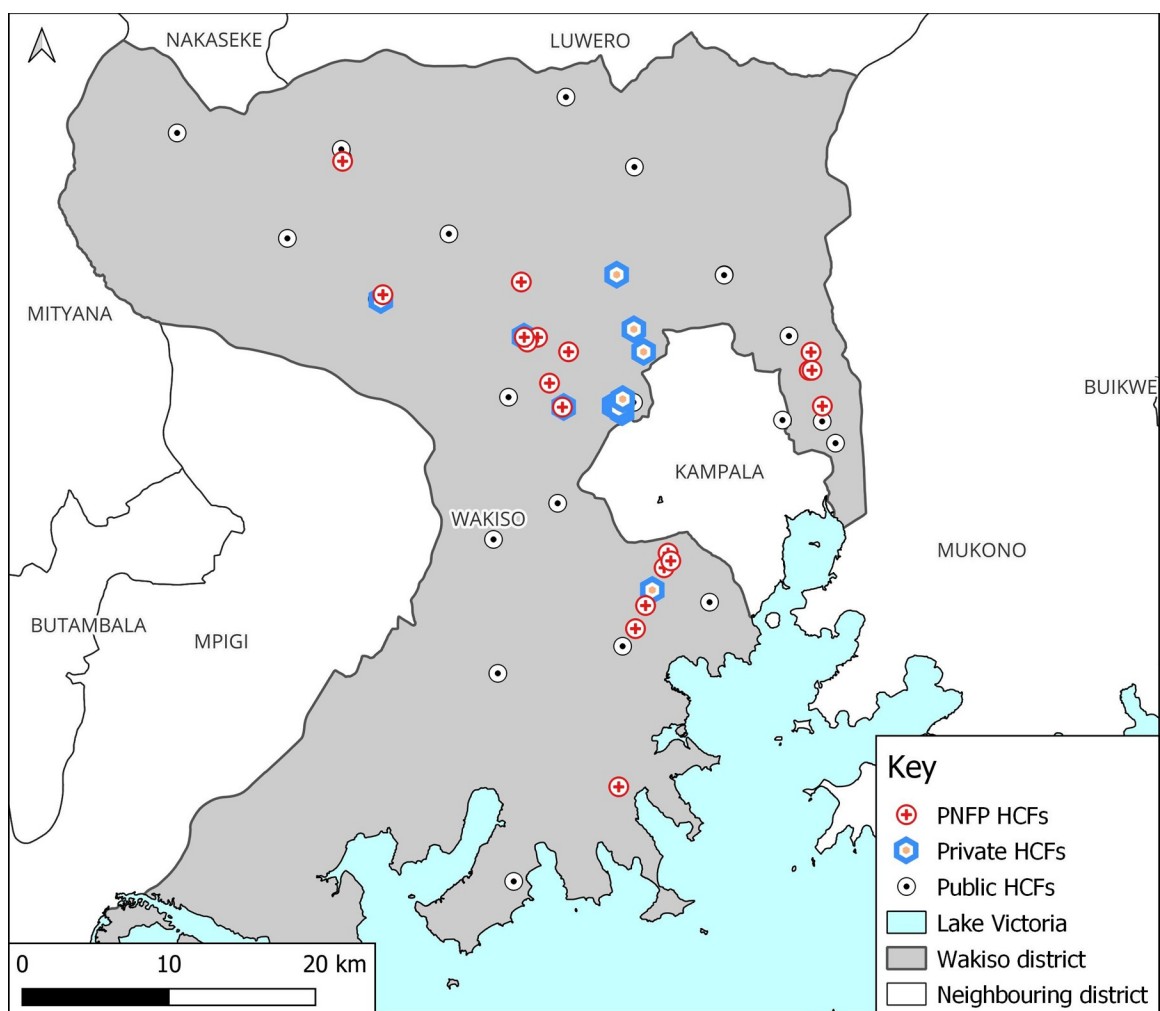

**Fig 2. Map of Wakiso district showing ownership status of the sampled healthcare facilities.**

hospitals, 17 health centre IVs and 34 IIIs were selected. Although the HCF inventory indicated a total of 477 private HCFs, only 47 met the inclusion criteria of providing high-risk medical interventions.

Upon selection of the HCFs, data were collected using the Kobo Collect mobile application which was installed on Android-enabled mobile phones and tablets, and preloaded with an electronic structured questionnaire. Research Assistants were required to upload the data onto the cloud server daily to avoid any risk of losing it. Upon completion of the survey, data were downloaded in the MS EXCEL format, checked for consistency, and cleaned. Afterwards, data were exported to STATA version 14.0 for analysis. Cross tabulations based on the level of the HCF, ownership, and location were done and are presented in the current paper. A Chi square ($\chi^2$) test was performed to establish the relationship between HCF characteristics and availability of hepatitis B related services and supplies. Our intention was to illustrate the availability and distribution of HBV prevention services across the different HCF characteristics and thus we did not conduct multivariable analyses.

## Mapping of HBV prevention services

The electronic questionnaire that was used to gather data on service availability also picked Geographical Positioning Systems (GPS) coordinates, i.e., northings and eastings. Location data were stored on the cloud server hosted by Kobo collect, together with attribute data about the HCFs. The data set was later downloaded as a MS EXCEL file, checked for inconsistencies, and cleared of any errors. The data file was then saved as a comma delimited text file and imported as a layer into a GIS environment using ArcGIS (version 10.1). The point plots were overlaid onto administrative boundaries of Wakiso district and its environs, obtained from the Ugandan Bureau of Statistics (UBOS). Interpolation was done using Kernel density estimation to produce a heatmap showing clustering of healthcare facilities. In addition, maps showing the distribution of different attributes such as availability of HBV, and routine vaccination schedule for health workers were generated and exported as a jpeg.

## Quality control and quality assurance measures

Data were collected by 6 research assistants who were supervised by the principal investigator (TS). Before data collection, research assistants underwent a four days' training to familiarise themselves with the data collection protocol. As part of the training, the data collection tools were pretested at two primary HCFs in peri-urban areas of Kampala City. The pre-test HCFs had comparable characteristics as those in the study district. These characteristics included conducting high-risk medical interventions such as conducting high volume deliveries, which increases occupational exposure to viral hepatitis. Research assistants picked geocoordinates while at the center of the HCF and at an accuracy of ±5 metres. During data management, data were visually inspected to detect any errors including, the absence of data, positional accuracy of data, missing, and misplaced features, and checking for any outliers. To double check spatial accuracy, the locations were mapped as Keyhole Markup Language (KML) and visualized on a base map in google earth to verify location relative to other features at the HCFs.

## Ethics statement

The study was approved by the Makerere University School of Public Health Higher Degrees Research and Ethics Committee. Administrative clearance was sought from the Wakiso district Local government and the management of the selected HCFs. Written informed consent was also obtained from the study respondents before interviews or observations could happen. All informed consent discussions were done in English since all health workers were conversant with the language.

## Results

### Background characteristics of health facilities

Out of the 55 facilities visited, most 61.8% (34/55) were health centres IIIs and 47.3% (26/55) were privately (PFP) owned. Most of the HCFs visited were located in the urban setting 65.4% (36/55) and Busiro North sub-district had the highest number 29.1% (16/55) of the HCFs visited (Table 1). Location data indicated high clustering of HCFs around Kampala, Uganda's capital (Fig 3).

**Table 1. Background characteristics of study healthcare facilities.**

| Variables | Category | Frequency (N = 55) | Percentage (%) |
|---|---|---|---|
| Level of health facility | Health Centre III | 34 | 61.8 |
| | Health Centre IV | 17 | 30.9 |
| | Hospital | 04 | 7.3 |
| Health facility Ownership | Public | 23 | 41.8 |
| | Private for-profit | 26 | 47.3 |
| | Private-not-for-profit | 06 | 10.9 |
| Location of the facility | Urban | 36 | 65.4 |
| | Rural | 19 | 34.6 |
| Name of the Health sub-district | Busiro North | 16 | 29.1 |
| | Busiro East | 7 | 12.7 |
| | Busiro south | 5 | 9.1 |
| | Kyadondo North | 10 | 18.2 |
| | Kyadondo south | 06 | 10.9 |
| | Kyadondo East | 10 | 18.2 |
| | Entebbe municipality | 01 | 1.8 |

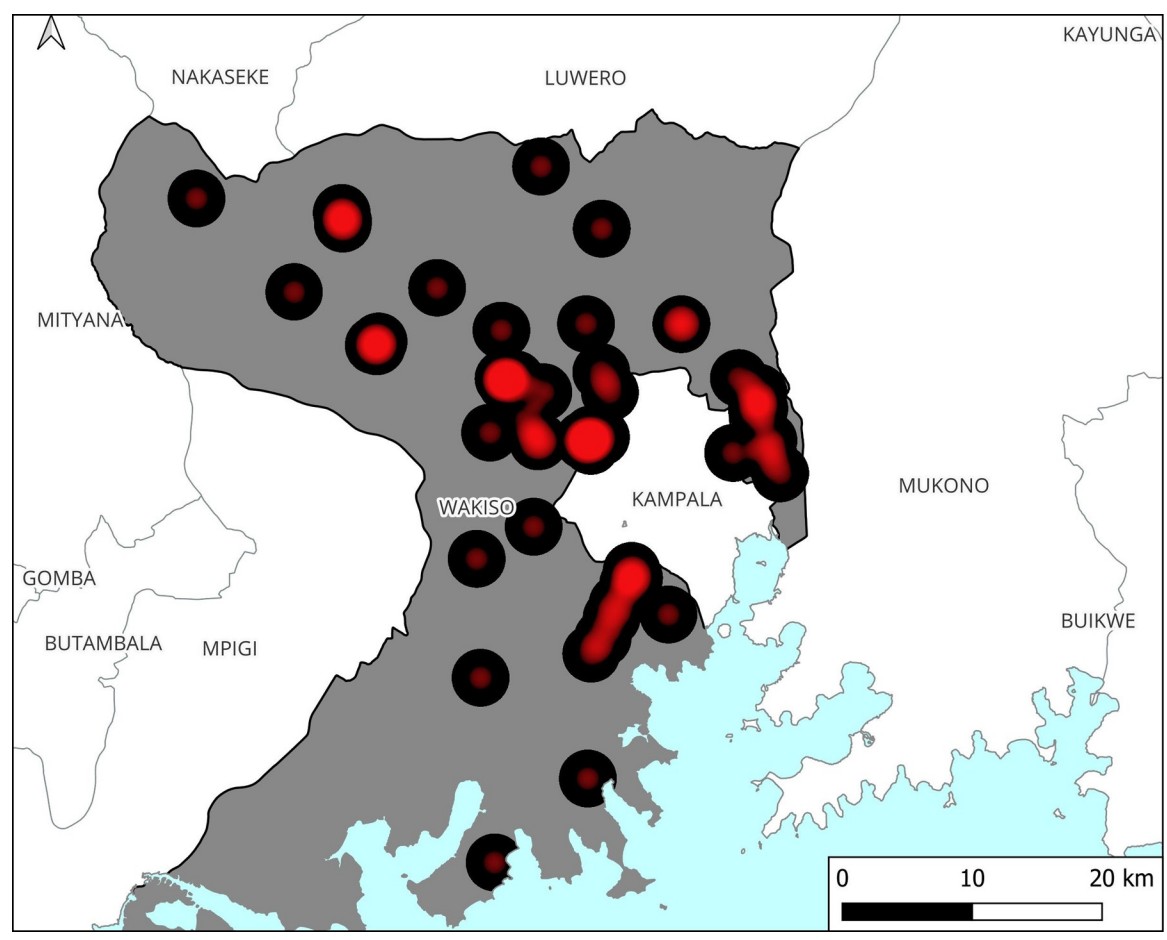

**Fig 3. A heat map showing clustering of healthcare facilities in Wakiso district.**

### Availability and distribution of Hepatitis B vaccine doses, testing kits and trained personnel

Only 29% (16) of the 55 HCFs reported having received hepatitis B vaccine doses in the last 12 months. At the time of the study, the hepatitis B vaccine was available in only 27.3% (15/55) of the facilities; mostly in HC IVs 52.9% (9/17) and lowest in HC IIIs 11.8% (4/34). More urban HCFs, 93.3% (14/15) had hepatitis B vaccine available at the time of the study than rural ones 6.7% (1/14). Most HCFs 60.0% (33/55) had testing kits among which all hospitals and PNFP facilities had hepatitis B testing kits at their facilities. More urban HCFs, 72.2% (26/36) had testing kits than rural ones 36.8% (7/19). Although 30.9% (17/55) of the HCFs had personnel trained on Hepatitis B management; only 1 hospital 25.0% (1/4) and 3 13.0% (3/23) public facilities had their personnel trained on hepatitis B management. Infection control promotion materials were found in 89.1% (49/55) of the facilities; 91.2% (31/34) in HC IIIs, 82.4% (14/17) in HCIV and all hospitals. Nearly three quarters 65.6% (36/55) of the HCFs had post-exposure prophylaxis (PEP) guidelines in case of accidental injuries; this was lowest in public facilities 65.2% (15/23). Almost all facilities 96.4% (53/55) had handwashing facilities (Table 2). The distribution of HCFs with HBV testing services and Vaccine doses is shown in Figs 4 and 5 respectively.

### Availability and distribution of routine vaccination services

Only 29.1% (16/55) of the HCFs surveyed had a routine vaccination schedule. The majority of the HCFs that had a routine vaccination schedule were clustered in the same area, with just a few dispersed outwards (Fig 6). The majority of the HCFs were also located just adjacent to Kampala, Uganda's Capital with more than half, 62.5% (10/16) in the urban areas and over a third, 37.5% (6/16) in the rural areas. The distribution of HCFs with a routine vaccination schedule is shown in Fig 6.

### Readiness and availability of hepatitis B prevention services stratified by healthcare facility characteristics

There was a statistically significant association between level of HCF and receiving hepatitis B vaccine doses in the last 12 months (p = ≤0.001), having Hepatitis B vaccines at the time of the survey (p = 0.002), availability of hepatitis B testing services (p = 0.031), having hepatitis B testing kits in stock (p = 0.013), having reminders and/or job aids that promote the reduction and use of injections, safe administration of injections, or safe disposal of used injection equipment available at this facility (p = 0.003) and having color-coded waste bins to promote safe waste management (p = 0.043) (Table 2).

There was a statistically significant association between ownership of the HCF and availability of guidelines to follow in case healthcare providers are exposed to the hepatitis B infection (p≤0.001), availability of hepatitis B testing services (p≤0.001), availability of hepatitis B testing kits (p≤0.001), poor management of infectious waste (p = 0.007), and having someone trained in the management of hepatitis B (p = 0.049).

The location of HCF was associated with receiving hepatitis B vaccine doses in the last 12 months (p = 0.030), and availability of Hepatitis B vaccines (p = 0.010), testing services (p = 0.010) and Hepatitis B testing kits (p = 0.004). The location of the HCF was also associated with availability of guidelines to follow in case healthcare providers are exposed to the hepatitis B infection (p = 0.040) and presence of staff trained in hepatitis B testing (p = 0.050).

**Table 2. Readiness and availability of hepatitis B prevention services in Wakiso district, Central Uganda, stratified by level of the healthcare facility, ownership, and location.**

| Indicator | (N = 55) (%) | Health facility level N (%) | | | $\chi^2$ p-value | Ownership N (%) | | | $\chi^2$ p-value | Location N (%) | | $\chi^2$ p-value |
|---|---|---|---|---|---|---|---|---|---|---|---|---|
| | | HCIII (N = 34) | HCIV (N = 17) | Hospital (N = 4) | | PFP (N = 26) | PNFP (N = 6) | Public (N = 23) | | Rural (N = 19) | Urban (N = 36) | |
| HCF received Hep B vaccine doses in the last 12 months | 16 (29.1) | 03 (8.8) | 10 (58.8) | 03 (75.0) | **≤0.001** | 10 (38.5) | 02 (33.3) | 4 (17.4) | 0.261 | 2 (10.5) | 14 (38.9) | **0.030** |
| HCF has functional refrigerators for storing Hepatitis B vaccines | 49 (89.1) | 28 (82.4) | 17 (100) | 04 (100) | 0.125 | 23 (88.5) | 04 (66.7) | 22 (95.7) | 0.127 | 16 (84.2) | 33 (91.7) | 0.399 |
| HCF has cold chain technicians | 19 (34.6) | 08 (23.5) | 09 (52.9) | 02 (50.0) | 0.052 | 09 (34.6) | 02 (33.3) | 08 (34.8) | 0.998 | 04 (21.1) | 15 (41.7) | 0.126 |
| Cold chain technician (s) at the HCF were trained in temperature monitoring | 14 (25.4) | 07 (20.5) | 06 (35.2) | 01 (25.0) | 0.502 | 07 (26.9) | 02 (33.3) | 05 (21.7) | 0.520 | 04 (21.1) | 10 (27.7) | 0.179 |
| HCF had Hepatitis B vaccines at the time of the survey | 15 (27.3) | 04 (11.8) | 09 (52.9) | 02 (50.0) | **0.002** | 10 (38.5) | 02 (33.3) | 03 (13.0) | 0.129 | 01 (5.3) | 14 (38.9) | **0.010** |
| HCF has guidelines to follow in case healthcare providers are exposed to the hepatitis B infection | 25 (45.6) | 14 (41.2) | 08 (47.1) | 03 (75.0) | 0.432 | 17 (65.4) | 06 (100) | 02 (8.7) | **≤0.001** | 05 (26.3) | 20 (55.6) | **0.040** |
| Hepatitis B testing services are available at the HCF | 33 (60.0) | 16 (47.1) | 13 (76.5) | 04 (100) | **0.031** | 23 (88.5) | 06 (100) | 04 (17.4) | **≤0.001** | 07 (36.8) | 26 (72.2) | **0.010** |
| HCF has Hepatitis B testing kits in stock at the time of the survey | 29 (52.7) | 13 (38.2) | 12 (70.6) | 04 (100) | **0.013** | 21 (80.8) | 05 (83.3) | 03 (13.0) | **≤0.001** | 05 (26.3) | 24 (66.7) | **0.004** |
| HCF has a routine vaccination schedule for health care providers and other target groups | 16 (29.1) | 08 (23.5) | 05 (29.4) | 03 (75.0) | 0.101 | 07 (26.9) | 02 (33.3) | 07 (30.4) | 0.936 | 06 (31.6) | 10 (27.8) | 0.768 |
| HCF has any infection prevention and control promotion materials | 49 (89.1) | 31 (91.2) | 14 (82.4) | 04 (100) | 0.488 | 25 (96.2) | 05 (83.3) | 19 (82.6) | 0.282 | 16 (84.2) | 33 (91.7) | 0.399 |
| HCF has reminders and/or job aids posted that promote reducing the use of injections, safe administration of injections, or safe disposal of used injection equipment available at this facility | 44 (80.0) | 31 (91.2) | 10 (58.8) | 03 (75.0) | **0.003** | 21 (80.8) | 06 (100) | 17 (73.9) | 0.360 | 17 (89.5) | 27 (75.0) | 0.202 |
| HCF holds continuous medical education (CMEs) on hepatitis B prevention | 16 (29.1) | 08 (23.5) | 07 (41.2) | 01 (25.0) | 0.418 | 08 (30.8) | 02 (33.3) | 06 (26.1) | 0.910 | 06 (31.6) | 10 (27.8) | 0.768 |
| HCF has color-coded waste bins to promote safe waste management | 48 (87.3) | 32 (94.1) | 12 (70.6) | 04 (100) | **0.043** | 22 (84.6) | 06 (100) | 20 (89.6) | 0.594 | 18 (94.7) | 30 (83.3) | 0.228 |
| HCF has poorly managed infectious waste | 24 (43.6) | 12 (35.3) | 10 (58.8) | 02 (50.0) | 0.270 | 09 (34.6) | 0 (0.0) | 15 (65.2) | **0.007** | 09 (47.4) | 15 (41.7) | 0.685 |
| HCF has staff trained in hepatitis B testing | 43 (78.2) | 25 (73.5) | 14 (82.4) | 04 (100) | 0.423 | 22 (84.6) | 06 (100) | 15 (65.2) | 0.100 | 12 (63.2) | 31 (86.1) | **0.050** |
| HCF has someone trained in the management of hepatitis B | 17 (30.9) | 09 (26.5) | 07 (41.2) | 01 (25.0) | 0.544 | 11 (42.3) | 03 (50.0) | 03 (13.0) | **0.049** | 05 (26.3) | 12 (33.3) | 0.592 |
| HCF has an infection prevention and control focal person/ committee | 20 (36.4) | 12 (35.3) | 05 (29.4) | 03 (75.0) | 0.229 | 07 (26.9) | 04 (66.7) | 09 (39.1) | 0.177 | 05 (26.3) | 15 (41.7) | 0.260 |
| HCF has post-exposure prophylaxis guidelines in case of injuries | 36 (65.6) | 23 (67.7) | 09 (52.9) | 04 (100) | 0.186 | 16 (61.5) | 05 (83.3) | 15 (65.2) | 0.599 | 13 (68.4) | 23 (63.9) | 0.737 |
| HCF has handwashing facilities with soap | 46 (83.6) | 30 (88.2) | 12 (70.5) | 04 (100) | 0.098 | 26 (100) | 06 (100) | 14 (60.8) | 0.236 | 15 (78.9) | 31 (86.1) | 295 |

## Discussion

The main aim of this study was to assess the availability and distribution of Hepatitis B prevention services in Wakiso District, Central Uganda. This is important since is unmasks variation in service availability across a geographic space. Our analysis found that only 29% of the HCFs had hepatitis B vaccine supplies in the last 12 months, but the vaccine was available in only 27.3% of the HCFs at the time of the survey. We established that less than half of the HCFs had infection prevention and control (IPC) guidelines to follow when exposed to HBV infections,

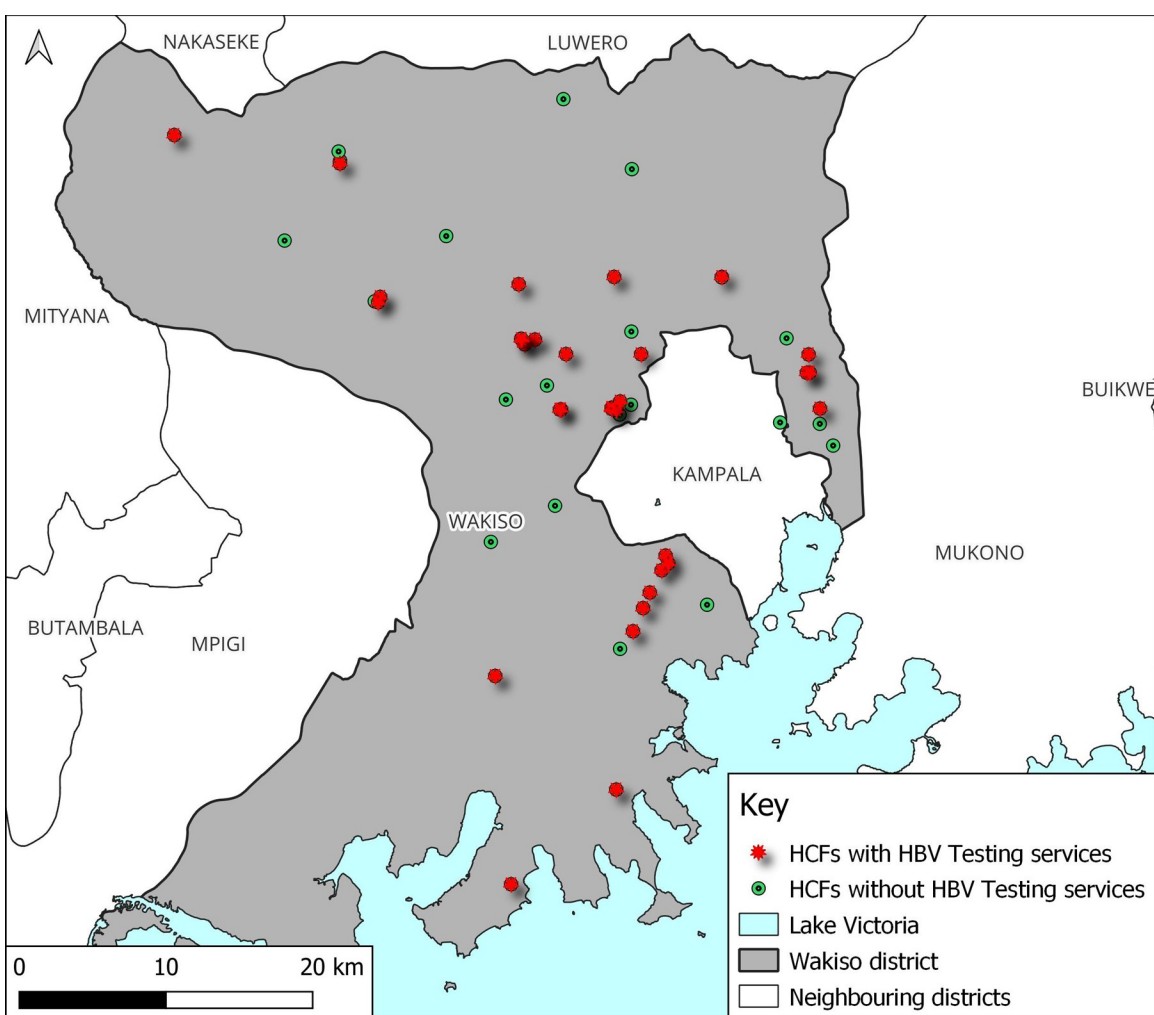

**Fig 4. A map of Wakiso district showing the distribution of healthcare facilities that offer hepatitis B testing services.**

and only two-thirds of the HCFs had PEP guidelines in case of injuries. The current study indicated that the majority of the HCFs offering routine HBV vaccination were clustered in the same area, with just a few dispersed outwards. Hepatitis B vaccination and testing services were mainly offered by health centre IVs and hospitals.

Analysis of the distribution of hepatitis B prevention services such as vaccination, screening, and testing, revealed that HCFs offering such services conformed to a nucleated pattern. HCFs were located close to each other. Many of these HCFs were located close to Kampala, Uganda's capital, and in municipalities and major town councils. The location or proximity of HCFs providing Hepatitis B prevention services to the more urbanised areas is not surprising. Urbanised areas are more likely to have a higher demand for services compared to the less urbanised or rural settings. There is also evidence that urban dwellers are more likely to be knowledgeable about their health, and thus can demand services. However, this has implications for hepatitis B prevention. It is known that rural dwellers in Uganda have lower incomes compared to urban dwellers [32,33]. Therefore, the unequal distribution of hepatitis B prevention services is likely to limit their uptake.

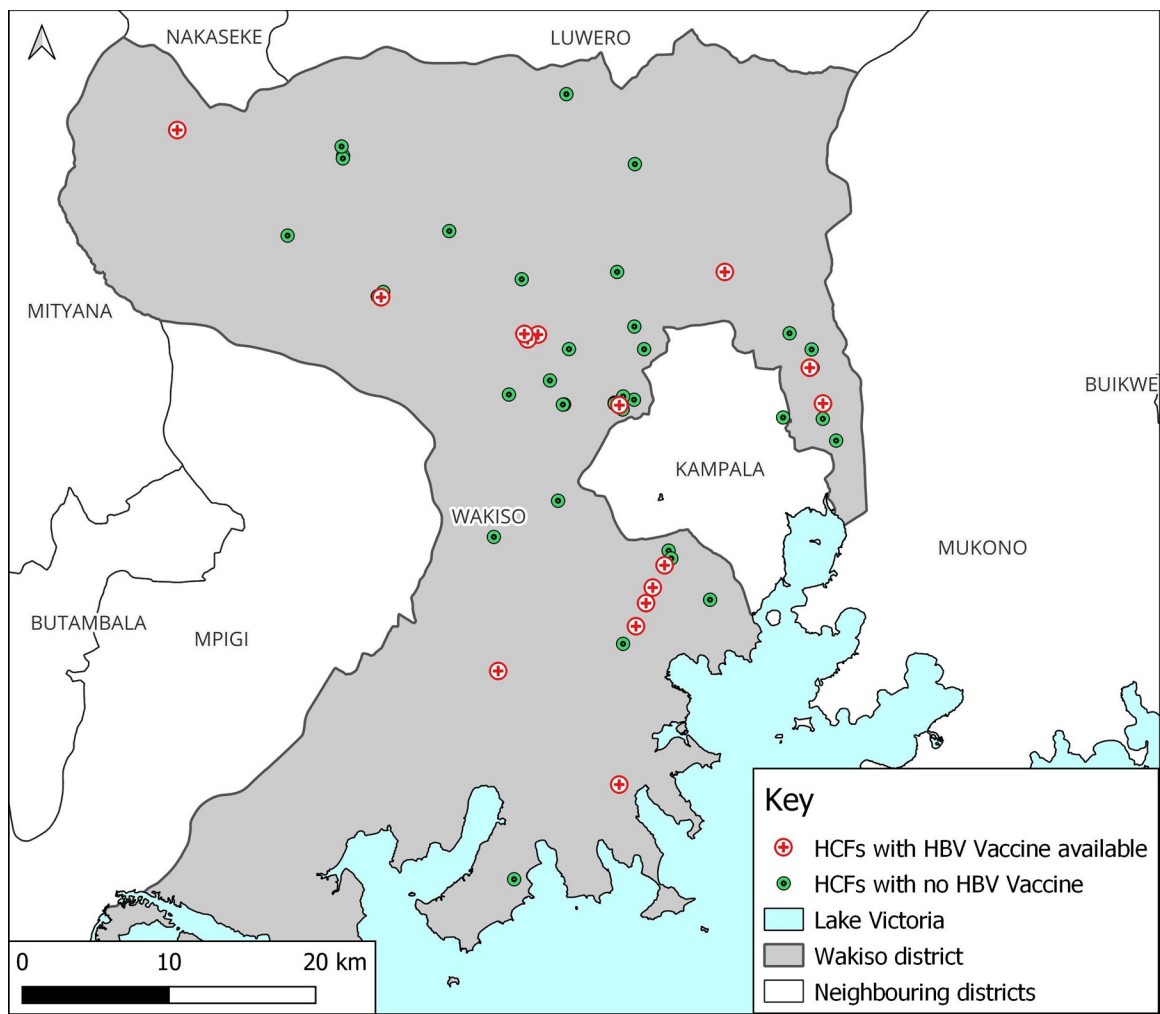

**Fig 5. A map showing healthcare facilities with hepatitis B vaccine in Wakiso district, Uganda.**

Our study revealed that only 29% of the HCFs reported having hepatitis B vaccine doses in stock in the last 12 months. Our findings indicate that less than a third of the HCFs had hepatitis B vaccines in stock during the study period (2019). The low proportion of HCFs with hepatitis B vaccine doses in stock during the study period may have been attributed to the low prevalence of the disease in the region. The central region, where Wakiso district is located had a lower prevalence (1.6%) of hepatitis B compared to other regions (4.6% in mid-north, 3.8% in West Nile, 4.4% in northeast among other regions) [34,35]. Due to the low prevalence of hepatitis B in the central region, Wakiso district was in 2015 not prioritised by the Ugandan Ministry of Health for mass hepatitis B prevention services including screening, vaccination of adults, and testing [35]. The low prevalence and consequently low prioritisation of hepatitis B prevention services by the Ugandan Ministry of Health may have driven the availability and distribution of hepatitis B services in HCFs in Wakiso district. Besides, the low demand for hepatitis B prevention services, which resulted from limited knowledge and a negative attitude towards hepatitis B prevention among healthcare providers in Wakiso district, as earlier reported by Ssekamatte, Isunju [31] could also explain the limited coverage in HCFs. The study reported that 42.2% of healthcare providers exhibited poor knowledge of HBV infection

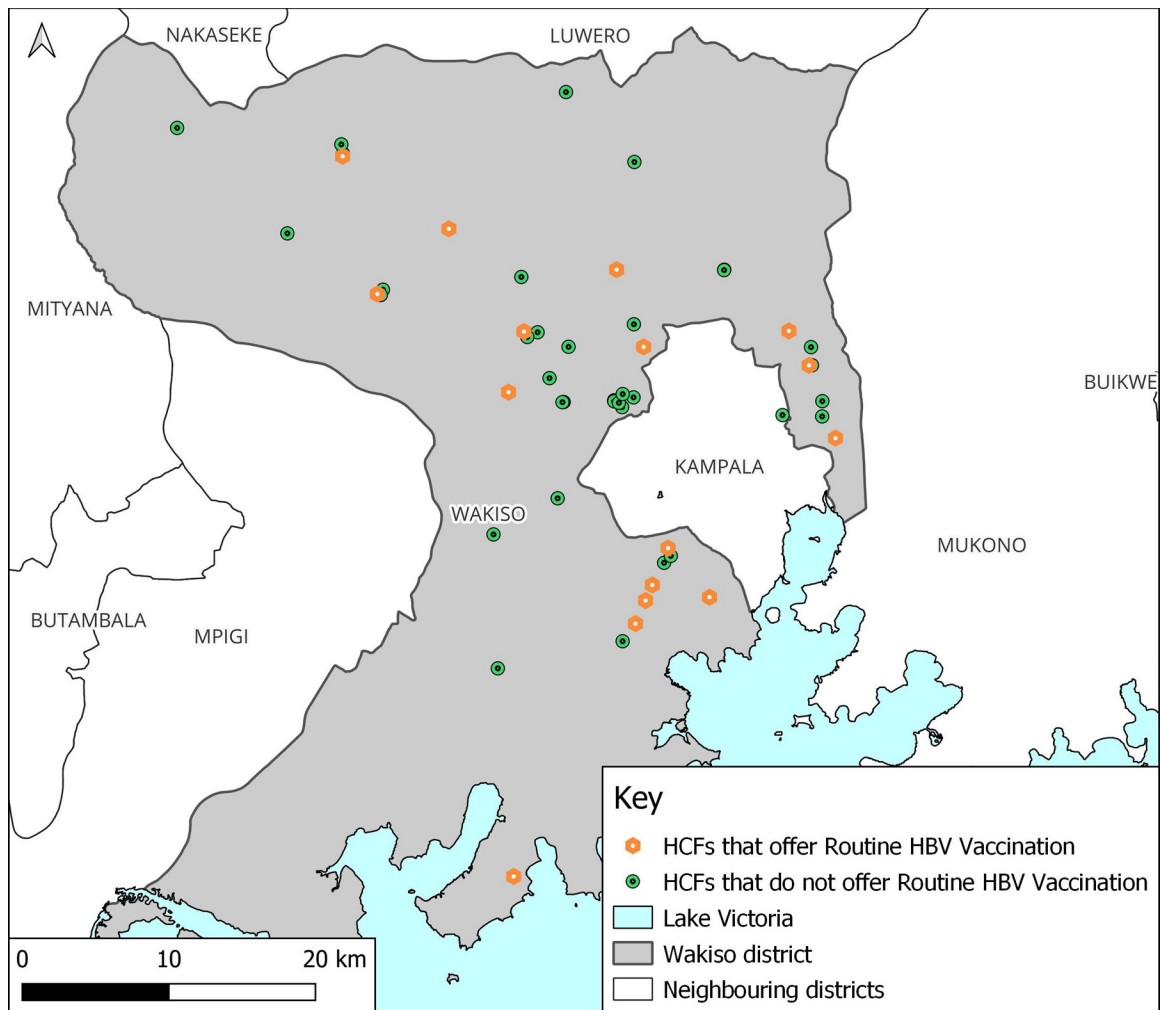

**Fig 6. A map of Wakiso district showing healthcare facilities with a routine vaccination schedule for healthcare providers and other high-risk groups.**

transmission and prevention, 41.8% had a negative attitude while 41.5% exhibited poor hepatitis B prevention practices. There is evidence that unfavourable knowledge, attitude, and practices can affect the demand and supply of healthcare services [36].

A higher proportion of PNFP and PFP HCFs had hepatitis B prevention services such as screening and testing compared to the public HCFs. This is not surprising given the fact that procurement of the vaccine greatly depended on the demand, and that the vaccine was paid for by the clients. Therefore, the provision of hepatitis B vaccination services such as testing and vaccination was seen as a business from which PNFP and PFP HCFs would make profits. There is evidence of HCFs providing services only if they can make a profit [37]. The low proportion of public HCFs reporting having hepatitis B vaccines in stock in the last 12 months could be attributed to the fact that Wakiso district was not in a high-burden region, and thus, not prioritised. The Ugandan Ministry of Health rolled out free vaccinations in the country starting with regions of high prevalence that did not include Wakiso district. Thus public/government HCFs in low burden areas did not have the vaccine readily available [35].

Less than half of the HCFs in our study setting had guidelines patients and healthcare providers could follow when exposed to the hepatitis B virus. In addition, about two-thirds of the HCFs had PEP guidelines in case of injuries. The low proportion of HCFs with infection prevention and control guidelines could have been due to the limited investment in hepatitis B prevention and control to support development and distribution of guidelines, policies and standards to HCFs. Limited funding is widely reported to impact the provision of healthcare services, and the development and implementation of guidelines and policies [38].

Less than a third of the study HCF held continuing medical education sessions (CMEs) on hepatitis B prevention. The CMEs helps healthcare providers to maintain, develop, or increase the knowledge and skills about the provision of healthcare. This is a considerably low proportion given the public health importance of hepatitis B. The low proportion of HCFs holding CMEs may have resulted from the fact that less than a third of the HCF had a healthcare provider trained in the management of hepatitis B, despite more than three quarters being trained in hepatitis B testing. The fact that only a few HCFs had healthcare providers trained in hepatitis B management indicates a knowledge gap among healthcare providers in HCFs in Wakiso district. Therefore, our study recommends the need to train healthcare providers, particularly those in health centre IIIs and public HCFs, irrespective of the location of HCF.

## Strengths and limitations

This is one of the few studies that has so far established the distribution of Hepatitis B prevention services. It is also one of the few studies that has used the SARA framework to establish the distribution and availability of HBV prevention services across HCF characteristics. The study mapped the geographic distribution and availability of HBV prevention services which makes it easy for the different stakeholders to interpret [39]. Despite these strengths, our study also had some limitations. The study measured few variables which made it difficult to establish factors that affect service availability. Additionally, the data did not fulfil the assumptions for multivariate analysis which made it difficult to control for confounding.

## Conclusions

Our study revealed an uneven distribution of hepatitis B services. The majority of the hepatitis B services were clustered around major towns, municipalities, and Kampala City, with just a few sparsely located in rural areas. Regarding the availability of hepatitis B prevention services, less than a third of the HCFs had hepatitis B vaccine doses in stock in the last 12 months, less than a third had a routine vaccination schedule for the health care providers while slightly more than half offered hepatitis B testing services. The proportion of HCFs offering hepatitis B services was higher among urban HCFs, PFP HCFs, and hospitals. This calls for the extension of hepatitis B prevention services to rural, public and PNFP HCFs.

## Supporting information

**S1 Text. Data collection tool.**
(PDF)

**S1 Data. Dataset.**
(XLSX)

## Acknowledgments

We thank the Wakiso District Health office for granting the study team administrative clearance to conduct the survey. Our sincere thanks also go to the management/administration of the different healthcare facilities, without whom, this study would not have been accomplished. Finally, we appreciate the research assistants for their role in undertaking the survey from which the current manuscript has been written.

## Author Contributions

**Conceptualization:** Tonny Ssekamatte, John Bosco Isunju, Rebecca Nuwematsiko, Winnifred K. Kansiime, Naume Muyanga, Justine N. Bukenya, Richard K. Mugambe.

**Formal analysis:** Solomon Tsebeni Wafula, Rebecca Nuwematsiko, Doreen Nakalembe.

**Methodology:** John Bosco Isunju, Aisha Nalugya, Naume Muyanga, Joana Nakiggala, Justine N. Bukenya, Richard K. Mugambe.

**Project administration:** John Bosco Isunju, Aisha Nalugya, Rebecca Nuwematsiko, Doreen Nakalembe, Winnifred K. Kansiime, Richard K. Mugambe.

**Resources:** Tonny Ssekamatte.

**Supervision:** Tonny Ssekamatte, Aisha Nalugya, Rebecca Nuwematsiko, Doreen Nakalembe, Winnifred K. Kansiime, Naume Muyanga, Joana Nakiggala, Justine N. Bukenya.

**Validation:** Justine N. Bukenya.

**Writing – original draft:** Tonny Ssekamatte, John Bosco Isunju, Aisha Nalugya, Solomon Tsebeni Wafula, Rebecca Nuwematsiko, Doreen Nakalembe, Winnifred K. Kansiime, Naume Muyanga, Joana Nakiggala, Justine N. Bukenya, Richard K. Mugambe.

**Writing – review & editing:** Tonny Ssekamatte, John Bosco Isunju, Aisha Nalugya, Solomon Tsebeni Wafula, Rebecca Nuwematsiko, Doreen Nakalembe, Naume Muyanga, Joana Nakiggala, Justine N. Bukenya, Richard K. Mugambe.

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
