## [Decision Letter · Decision Letter 0]

25 Jul 2022

PGPH-D-22-00638

Geospatial distribution of Hepatitis B prevention services in Wakiso District, Central Uganda

Dear Dr. Ssekamatte,

Thank you for submitting your manuscript to PLOS Global Public Health. After careful consideration, we feel that it has merit but does not fully meet PLOS Global Public Health’s publication criteria as it currently stands. Therefore, we invite you to submit a revised version of the manuscript that addresses the points raised during the review process.

EDITOR: Please insert comments here and delete this placeholder text when finished. Be sure to:

Indicate which changes you require for acceptance versus which changes you recommendAddress any conflicts between the reviews so that it's clear which advice the authors should followProvide specific feedback from your evaluation of the manuscript

Please ensure that your decision is justified on PLOS Global Public Health’s publication criteria and not, for example, on novelty or perceived impact.

We look forward to receiving your revised manuscript.

Kind regards,

Abdisalan Mohamed Noor, Ph.D.

Section Editor

Journal Requirements:

Additional Editor Comments (if provided):

Reviewers' comments:

Reviewer's Responses to Questions

**Comments to the Author**

1. Does this manuscript meet PLOS Global Public Health’s publication criteria? Is the manuscript technically sound, and do the data support the conclusions? The manuscript must describe methodologically and ethically rigorous research with conclusions that are appropriately drawn based on the data presented.

Reviewer #1: No

Reviewer #2: No

2. Has the statistical analysis been performed appropriately and rigorously?

Reviewer #1: No

Reviewer #2: No

3. Have the authors made all data underlying the findings in their manuscript fully available (please refer to the Data Availability Statement at the start of the manuscript PDF file)?

Reviewer #1: Yes

Reviewer #2: No

4. Is the manuscript presented in an intelligible fashion and written in standard English?

Reviewer #1: Yes

Reviewer #2: Yes

5. Review Comments to the Author

Reviewer #1: Geospatial distribution of Hepatitis B prevention services in Wakiso District, Central Uganda

This is a cross-sectional descriptive study conducted in 55 healthcare facilities to assess service availability of hepatitis B vaccine and testing services in Wakiso District, Central Uganda. Based on the assessment the hepatitis B vaccine was available in only 27.3 %, while testing services were available 60% the facilities. Based on descriptive analysis, level of health care facility and location showed significant association with availability of hepatitis B vaccine. While hepatitis B testing services was associated with level of healthcare facility, ownership and location. This is an important area to be studied and the results of this study inform the Wakiso District. Nonetheless the number of districts included the study is very small, the analysis used is not in congruent with the title, even so the authors only did a univariate analysis with our controlling the potential confound effect of the variables one another.

Specific comments

• The title ‘Geospatial distribution of Hepatitis B prevention services in Wakiso District, Central Uganda’, is misleading as such there is no geospatial analysis conducted in this study. The only thing here is using the coordinates the location of the health facilities has been mapped. Better to remove the term ‘geospatial’ or conduct appropriate rigorous analysis.

• What is the key message of this paper ‘Healthcare facilities offering vaccination and testing services were mostly in urban healthcare facilities’, is that not clear that facilities in urban area are better off in-service provision overall, not particular to this analysis?

• The assertion ‘close to Kampala, Uganda’s capital’ is not supported by the available data. Were there measurements in terms of distance of the health facilities from Kampala? It seems most of the health facilities included in the study are close to Kampala.

• There is no clear justification of selection of 55 (9%) health facilities out of the 589 HCFs in the district. Does this give enough sample size to make the conclusion the authors and making in the manuscript? The allocation of the sample does not look proportional for example, although 477(81%) health facilities were owned privately only few were sampled (47%).

• The number of hospitals included in the study differs between the text (line 172, indicates 6) and the table 1 (indicates 4).

• The variables measured are very limited. The authors should have developed a conceptual framework with clear indication of proximal and distal factors which affect the service availability.

• The analysis done is univariate analysis, it is worth doing multiple logistic regression analysis to assess factors independently associated with the services.

• Table 2 needs rework, the outcome of the analysis should be availability of the hepatitis B vaccine and testing services. Some of the variables include could be explanatory variables such as availability of refrigerator, presence of trained man power etc.

Reviewer #2: 1. Is it that the distribution of care and treatment services for hepatitis B prevention only unknown in Wakiso district in the whole country as alluded? What makes Wakiso district a unique focus in the entire country relative to all the other districts in this study?

2. Under study setting, a map of Wakiso district with its main features and its position relative to Uganda would be useful in simplifying the details in this section and orienting the reader who is not familiar with Uganda.

3. As the questionnaire were informed to some extents by SARA, the authors should comment whether there is a SARA or availability of other surveys that have collected similar information for hepatitis B in Uganda.

4. Provide an improved description of the checks that were done to verify the quality of the GPS data. At the moment only a summary is given and does not detail the actual checks that we conducted such as accuracy of the location, GPS coordinates being next to a structure etc.

5. Major: The section on Geospatial distribution requires more work. Creating a shapefile (vector file) from the collected GPS data is not necessary to describe as it’s a routine process. However, what could be useful to describe if any geoprocessing was done to the collected variables beyond visualizing them on map so that the section can stand on its own. At the moment its only describing how the GPS data was collected and creation of shapefile which can be summarized in 2-3 lines. If the sampling was representative and random, clustering analysis, interpolation among other geospatial techniques would have been useful in providing subnational metrics in this district.

6. My second major concern is to what extent can the results of purposively selected facilities be deemed to show the geospatial distribution of outcomes and proportions in a district? Would these maps and statistics not be misleading to the audience? The results are even further stratified by rural/urban, private/public facilities from a purposive survey. Since the survey is random would these proportions and maps not be biased?

7. I would suggestion removing details about the geospatial patterns in the manuscript and in the title since no clustering analysis was done. This is because only the location of the health facilities was collected and visualized in map together with the associated attributes from non-random survey.

6. PLOS authors have the option to publish the peer review history of their article (what does this mean?). If published, this will include your full peer review and any attached files.

**Do you want your identity to be public for this peer review?** For information about this choice, including consent withdrawal, please see our Privacy Policy.

Reviewer #1: No

Reviewer #2: No

---

## [Decision Letter · Decision Letter 1]

15 May 2023

PGPH-D-22-00638R1

Distribution of Hepatitis B prevention services in Wakiso District, Central Uganda

Dear Dr. Ssekamatte,

Thank you for submitting your manuscript to PLOS Global Public Health. After careful consideration, we feel that it has merit but does not fully meet PLOS Global Public Health’s publication criteria as it currently stands. Therefore, we invite you to submit a revised version of the manuscript that addresses the points raised during the review process.

We look forward to receiving your revised manuscript.

Kind regards,

Max Carlos Ramírez-Soto, BSc, MPH, FRSPH, MACE

Academic Editor

Journal Requirements:

Additional Editor Comments (if provided):

Reviewers' comments:

Reviewer's Responses to Questions

**Comments to the Author**

1. If the authors have adequately addressed your comments raised in a previous round of review and you feel that this manuscript is now acceptable for publication, you may indicate that here to bypass the “Comments to the Author” section, enter your conflict of interest statement in the “Confidential to Editor” section, and submit your "Accept" recommendation.

Reviewer #2: (No Response)

Reviewer #3: All comments have been addressed

2. Does this manuscript meet PLOS Global Public Health’s publication criteria? Is the manuscript technically sound, and do the data support the conclusions? The manuscript must describe methodologically and ethically rigorous research with conclusions that are appropriately drawn based on the data presented.

Reviewer #2: Partly

Reviewer #3: Yes

3. Has the statistical analysis been performed appropriately and rigorously?

Reviewer #2: No

Reviewer #3: Yes

4. Have the authors made all data underlying the findings in their manuscript fully available (please refer to the Data Availability Statement at the start of the manuscript PDF file)?

Reviewer #2: Yes

Reviewer #3: Yes

5. Is the manuscript presented in an intelligible fashion and written in standard English?

Reviewer #2: Yes

Reviewer #3: Yes

6. Review Comments to the Author

Reviewer #2: "We did not conduct multivariate analysis to control for confounding because the data did not fulfil the required assumptions i.e., multicollinearity, and independence of observations (29, 30)." This does not justify why multivariate analysis was not done. However, if the intention was to illustrate the availability and distribution of HBV prevention services across the different healthcare facility characteristics then the details on multivariuate should be omitted.

One of my previous comment was only adressed partly: "Comment: My second major concern is to what extent can the results of purposively selected facilities be deemed to show the distribution of outcomes and proportions in a district?

Would these maps and statistics not be misleading to the audience? The results are even further stratified by rural/urban, private/public facilities from a purposive survey. Since the survey is random would these proportions and maps not be biased?

Reviewer #3: The article seems pertinent to the objectives that were set to determine. It would be convenient if they could specify the on-site verification of the information obtained through the web questionnaire. Also in having more information on the availability of vaccines, human resources and determinants for the lack of access to vaccination, which is the most cost-effective strategy to avoid chronic HBV infection and its sequelae such as cancer and cirrhosis.

7. PLOS authors have the option to publish the peer review history of their article (what does this mean?). If published, this will include your full peer review and any attached files.

**Do you want your identity to be public for this peer review?** For information about this choice, including consent withdrawal, please see our Privacy Policy.

Reviewer #2: No

Reviewer #3: **Yes: **CESAR CABEZAS

---

## [Decision Letter · Decision Letter 2]

8 Aug 2023

Distribution of Hepatitis B prevention services in Wakiso District, Central Uganda

PGPH-D-22-00638R2

Dear Tonny,

We are pleased to inform you that your manuscript 'Distribution of Hepatitis B prevention services in Wakiso District, Central Uganda' has been provisionally accepted for publication in PLOS Global Public Health.

Best regards,

Collins Otieno Asweto, PhD

Academic Editor

Reviewer Comments (if any, and for reference):

Reviewer's Responses to Questions

**Comments to the Author**

1. If the authors have adequately addressed your comments raised in a previous round of review and you feel that this manuscript is now acceptable for publication, you may indicate that here to bypass the “Comments to the Author” section, enter your conflict of interest statement in the “Confidential to Editor” section, and submit your "Accept" recommendation.

Reviewer #2: All comments have been addressed

Reviewer #3: All comments have been addressed

2. Does this manuscript meet PLOS Global Public Health’s publication criteria? Is the manuscript technically sound, and do the data support the conclusions? The manuscript must describe methodologically and ethically rigorous research with conclusions that are appropriately drawn based on the data presented.

Reviewer #2: Yes

Reviewer #3: Yes

3. Has the statistical analysis been performed appropriately and rigorously?

Reviewer #2: Yes

Reviewer #3: Yes

4. Have the authors made all data underlying the findings in their manuscript fully available (please refer to the Data Availability Statement at the start of the manuscript PDF file)?

Reviewer #2: Yes

Reviewer #3: Yes

5. Is the manuscript presented in an intelligible fashion and written in standard English?

Reviewer #2: Yes

Reviewer #3: Yes

6. Review Comments to the Author

Reviewer #2: (No Response)

Reviewer #3: Suggestions made are included.

7. PLOS authors have the option to publish the peer review history of their article (what does this mean?). If published, this will include your full peer review and any attached files.

**Do you want your identity to be public for this peer review?** For information about this choice, including consent withdrawal, please see our Privacy Policy.

Reviewer #2: No

Reviewer #3: **Yes: **CESAR CABEZAS
